# Nanoparticles and Radioisotopes: A Long Story in a Nutshell

**DOI:** 10.3390/pharmaceutics14102024

**Published:** 2022-09-23

**Authors:** Giulia Poletto, Laura Evangelista, Francesca Venturini, Fabiana Gramegna, Flavio Seno, Stefano Moro, Roberto Vettor, Nicola Realdon, Diego Cecchin

**Affiliations:** 1Department of Medicine (DIMED), Nuclear Medicine Unit, Padova University Hospital, 35128 Padova, Italy; 2Pharmacy Unit, Padova University Hospital, 35128 Padova, Italy; 3INFN, National Laboratories of Legnaro, University of Padova, 35131 Padova, Italy; 4Department of Physic and Astronomy, INFN Section, University of Padova, 35121 Padova, Italy; 5Department of Pharmaceutical and Pharmacological Sciences, University of Padova, 35131 Padova, Italy

**Keywords:** radiopharmaceuticals, drug delivery, nanoparticles

## Abstract

The purpose of this narrative review was to assess the use of nanoparticles (NPs) to deliver radionuclides to targets, focusing on systems that have been tested in pre-clinical and, when available, clinical settings. A literature search was conducted in PubMed and Web of Science databases using the following terms: “radionuclides” AND “liposomes” or “PLGA nanoparticles” or “gold nanoparticles” or “iron oxide nanoparticles” or “silica nanoparticles” or “micelles” or “dendrimers”. No filters were applied, apart from a minimum limit of 10 patients enrolled for clinical studies. Data from some significant studies from pre-clinical and clinical settings were retrieved, and we briefly describe the information available. All the selected seven classes of nanoparticles were highly tested in clinical trials, but they all present many drawbacks. Liposomes are the only ones that have been tested for clinical applications, though they have never been commercialized. In conclusion, the application of NPs for imaging has been the object of much interest over the years, albeit mainly in pre-clinical settings. Thus, we think that, based on the current state, radiolabeled NPs must be investigated longer before finding their place in nuclear medicine.

## 1. Introduction

One method of drug delivery to lead molecules to their target is to use nano-sized carriers called nanoparticles (NPs). Different NPs can be used for this purpose, including polymeric NPs, liposomal carriers, dendrimers, magnetic iron oxide NPs, carbon nanotubes, and inorganic metal-based nanoformulations [1]. To ensure a homogenous delivery, NPs must be of suitable size and shape and have a suitable surface charge [2]. Different cargoes, such as hydrophilic or amphiphilic drugs, genes [3], and radionuclides, can be delivered with NPs, thereby increasing the target to non-target ratio [3,4].

Many advantages can derive from using NPs to deliver radionuclides. One concerns a reduction in the radionuclide’s toxicity: using radioactive polymeric NPs can limit the damage to healthy tissues due to a nonspecific delivery [1]. Another in the field of diagnostics lies in that using radioactive polymeric NPs can reduce the radiation dose needed to perform a scan and/or the time it takes to perform [1]. Third, NPs can carry high payloads of radionuclides, which could lead to an increase in their activity, thus enhancing the dose delivered to the tumor [1]. At least, in theory, this mode of delivery can therefore be very useful in the diagnosis and the treatment of patients or, in other words, in theragnostics [1].

In the last two decades, many studies have been performed on the delivery of numerous radionuclides using NPs. Several methods have been developed, such as direct or indirect surface labeling, radionuclide incorporation, and surface engineering (e.g., PEGylation) [1]. Despite promising early results, many NPs have never gone beyond the pre-clinical phase, and few delivery strategies have been tested in clinical trials.

This narrative review aimed to understand which is state of the art in the use of NPs in nuclear medicine by considering liposomes, silica NPs, gold NPs, PLGA NPs, iron oxide NPs, micelles, and dendrimers (Figure 1) to answer the question: will NPs, for future, be the delivery system of choice for radionuclides?

## 2. Materials and Methods

A literature search was conducted in PubMed and Web of Science databases using the following terms: “radionuclides” AND “liposomes” or “PLGA nanoparticles” or “gold nanoparticles” or “iron oxide nanoparticles” or “silica nanoparticles” or “micelles” or “dendrimers”. No filters were used, apart from a minimum limit of 10 patients enrolled for clinical studies. The following information was recorded for all studies: year of publication; radionuclide used; type of study (pre-clinical or clinical); whether a chemotherapeutic and the radionuclide were present simultaneously on the NP; the impact of the nanocarrier’s composition on drug delivery. For the clinical studies, the type of disease treated and the number of patients involved were also considered.

## 3. Results

### 3.1. Mechanism of Action

There are three possible pathways by which NPs can reach a tumor site. The first is *passive*, taking advantage of the enhanced permeation and retention (EPR) effect. Solid tumors are characterized by a vasculature full of “big fenestrations” that facilitate the passage of nano-sized particles and by poor lymphatic drainage that enhances the retention of NPs at the tumor site [5]. This passive targeting is hindered, however, by the rapid uptake of NPs in the liver and spleen, especially if they are administered intravenously [5]. To prolong their half-life, NPs can be coated with chains of polyethylene glycol (PEG) [5], which prevents their recognition and subsequent uptake by the reticuloendothelial system (RES). The second pathway is *active* when various agents with a high targeting affinity are applied to the surface of the NPs. These agents may be ligands or antibodies that usually target overexpressed extracellular or intracellular constituents of tumors [1,2,5].

However, active targeting has some limitations. Indeed, tumor accumulation is mainly associated with the EPR effect rather than the targeting agents. However, the agent can act only after the extravasation to improve the delivery system to the target. Nevertheless, the extravasation of NPs is associated with diverse conditions of the tumor environment, such as endothelial lining, tumor cell density, and matrix. Indeed, in some specific cases, the presence of a targeting agent cannot help the NPs reach the target. Finally, an increase in the immunogenicity of the NPs should be considered when binding targeting moieties on their surface [6]. The most important advantage derived from targeting agents is that they enhance the uptake of NP in the cancer cells; thus, they can be more accumulated in tumors than untargeted NPs [6].

The surface charge of NPs affects their biodistribution profile. This could make them interact with different biological environments, thus influencing cellular uptake. The most suitable interaction is the one that occurs between positively charged NPs and negatively charged cell membranes because it enhances the particles’ internalization [2]. The third pathway entails administering NPs directly to the tumor site via *intra-tumor injections* [5]. The NPs are thus placed directly where they are intended to act, partially avoiding biodistribution and pharmacokinetic issues.

### 3.2. Liposomes

Liposomes are nano-sized vesicles composed of lipids arranged in a bilayer surrounding an aqueous core [3,7]. It seems necessary to add an appropriate amount of cholesterol to the lipid bilayers to ensure their stability [8].

Liposomes are drug carriers that can encapsulate both hydrophilic drugs (thanks to their aqueous core) and hydrophobic drugs (trapped in the lipid bilayer). Their pharmacokinetic properties can be modified by changing the size and chemical composition of the lipid bilayer, the surface charge, and other characteristics (using PEG, for instance, or specific cell targeting agents) [3,4,7,8].

Among all NPs, liposomes are the most often used in clinical settings, especially for anticancer drug delivery [4]. In 1995, the Food and Drug Administration (FDA) approved a liposomal formulation of doxorubicin, called Doxil, for cancer treatment [7].

Liposomes can also be functionalized with a radionuclide. The two main radiolabeling techniques used are surface and intraliposomal labeling (Figure 2).

In the former, radionuclides are inserted in the lipid bilayer using a mechanism involving the radionuclide’s chelation by the phosphonate group [4,9,10]. Alternatively, an appropriate chelator can be attached to the phospholipid or the PEG chain on the liposome’s surface [4,11,12,13,14,15,16]. Different radionuclides require the use of different chelators, of course. If the radionuclide is non-metallic, labeling can be done using a covalent bond with PEGylated/non-PEGylated lipids [4,17,18]. An issue associated with the use of chelators for surface labeling lies in that the chelators may affect the biodistribution of the liposomes through cross-reaction with proteins in the blood [4].

With intraliposomal labeling, on the other hand, the radionuclide is incorporated inside the liposome’s core. Various strategies can be used to internalize the radionuclide, including *ionophore-chelator binding, unassisted loading, ionophore-drug binding*, and *remote loading*. Ionophores are molecules that allow metal ions to be transported across the lipid bilayer. *Ionophore-chelator binding* is the method most commonly used for intraliposomal labeling because, once the radionuclide has been transported inside the core, a stable complex is formed between the radionuclide and the chelator [4,19,20,21,22,23,24]. In *unassisted loading*, liposomes are filled with chelators that prompt the radionuclide’s internalization between the lipid bilayers [4,25]. The drawback of this technique is that the liposome must have already been filled with the chelator, so this method cannot be used for the radiolabeling of liposomal nanomedicines already in the market, such as Doxil [4]. In *ionophore-drug binding*, a drug chelates the radionuclide after its ionophore-mediated transportation across the lipid bilayer, so there is no need to use any type of chelator [4,26,27]. As for *remote loading*, this involves labeling preformed liposomes with radionuclides which create complexes with chelators or organic compounds outside the liposomal core [4,28,29,30,31,32,33]. Intraliposomal labeling improves the in-vivo stability of the radio-liposomes, partly by avoiding any interaction between the radionuclide and biological components and partly because there is no surface modification involved, so the physicochemical properties of the original liposome remain unaltered [4].

When choosing the radionuclide to use, its half-life should be considered carefully. Indeed radionuclides with a short half-life are poorly suited to the comparatively long time it takes for liposomes to reach tumors [33]. It is worth bearing in mind that liposomes have a half-life of about 10–20 h in mice and 30–90 h in humans. This makes it preferable to use radionuclides with a longer half-life for imaging purposes in both mice and humans [34], even at the expense of a higher effective dose (mSv) for the patient.

#### 3.2.1. Pre-clinical Aspects Regarding Liposomes

Many elements on the bilayer can influence the pharmacokinetic properties of liposomes, so pre-clinical studies are mandatory. The opportunity to radiolabel liposomes has proved helpful in such investigations because imaging techniques enable the visualization of liposome distribution in vivo (Table 1).

Liposomes’ distribution is influenced mainly by their size, lipid composition, quantity of cholesterol, and surface characteristics such as charge and hydrophilicity [8]. Half of the cholesterol in the lipid bilayer was demonstrated to be optimal [8], whereas the positive charge seemed to increase the liposome’s clearance from blood [37]. Even the radiolabeling technique can influence the pharmacokinetic properties of liposomes. Indeed, remote-loaded liposomes circulated longer than the surface-labeled ones, whereas the latter were less stable [34].

The presence of PEG on a liposome’s surface alters its distribution by extending its circulation time. However, the biodistribution of vesicles is influenced not only by the length of the PEG chain but also by the amount of PEG linked to the surface. After the administration of PEGylated liposomes, the organs receiving the highest dose of radiation were seen to be the spleen and liver, the walls of the stomach and lower large intestine, red marrow, and lungs, regardless of the mol% of PEG. Moreover, despite a higher tumor accumulation for liposomes containing a higher amount of PEG, those with a lesser amount were decreed as more suitable from a diagnostic standpoint [36].

All the above-mentioned studies invariably highlighted the high spleen and liver uptake of the liposomal NPs, despite their surface functionalization with PEG. Indeed, the presence of different amounts of PEG on the liposomes’ surface could only delay but not prevent spleen and liver uptake [36].

Even attaching a targeting agent to a liposome’s surface may enhance its efficient targeting of the tumor cell, though this was not always true [35]. Indeed, not only the targeting agent but all the NP’s features (e.g., the length of the PEG chain, the type of peptide used, and the amount of peptide load) should be considered as a factor influencing the receptor binding [35].

Even liposomal formulations can be radiolabeled. Thus, liposomes could become a non-invasive method for studying the kinetic properties of liposomal drugs and for selecting patients who would respond better to liposomal therapy [33]. Because of these promising results, liposomes have also been tested in patients.

Finally, liposomes were considered for their application in tumor therapy. This raises two issues, however. First, it is essential to verify the ^177^Lu liposomes’ stability, and second, patients suitable for therapy with ^177^Lu should be identified using pretreatment PET imaging. Thus, a study [36] tested the feasibility of simultaneously loading ^64^Cu and ^177^Lu in the same vesicle to obtain a potentially theragnostic agent. Results were promising. Liposomes seemed to ensure adequate radiation delivery to solid tumors, though the authors judged these results very promising but only preliminary.

#### 3.2.2. Clinical Aspects Regarding Liposomes

The use of radiolabeled liposomes was tested in clinical studies (Table 2) for imaging purposes only.

The first studies in the 1980s focused on the kinetic properties of these vesicles in patients with various tumors. Unfortunately, they did not all produce encouraging results using these NPs for radionuclide delivery. For instance, despite promising results in rat models, negatively charged ^99m^Tc liposomes injected in humans could not become localized in tumors [10].

On the other hand, by changing the composition of liposomes, i.e., making smaller and neutral vesicles, some studies [38,39] demonstrated the effectiveness of liposomes in detecting different types of tumors. Several unsuspected tumor areas were identified with those liposomes, too, and subsequently confirmed using other techniques. The overall sensitivity was 85% for individual sites, and the specificity was 96% [38,39]. Moreover, according to Man, these studies identify the heterogeneity of the EPR effect in humans [44].

Despite conflicting results, the above-mentioned studies were concordant in defining the PK of liposomes. After the injection, there was a high uptake of liposomes by the liver, spleen, and bone marrow [10,38,39]. Moreover, a study confirmed that neither the presence of PEG on liposomes’ surface could help them to escape uptake by the RES, although thanks to the PEG chains, the liposomes circulated for longer after injection, making the tumor clearly visible within 48–72 h [41].

As in pre-clinical applications, so too in clinical fields, radiolabeling had been used to tag liposomal formulations and study their accumulation in tumors and normal tissues. Liposomal doxorubicin radiolabeled with [^99m^Tc] DTPA was useful to test the status of the vasculature to identify patients who will respond better to treatment with doxorubicin [40]. [^99m^Tc]-labeled liposomal doxorubicin (^99m^Tc-LD) was used even to assess the deposition of these liposomes in tumors and their activity when combined with cisplatin for the treatment of malignant pleural mesothelioma. With this study, the authors could also identify which patients would respond better to chemotherapy based on liposomal doxorubicin and cisplatin [42]. Looking at these two studies, the EPR effect seems essential to the NP’s uptake and consequently for predicting response to therapy.

After the pre-clinical promising studies, ^64^Cu-MM-302 nano-liposomes [33] were examined for their practical application in humans [43]. The accumulation of ^64^Cu-MM-302 liposomes was seen to depend more on the characteristics of the lesion than on the NPs’ pharmacokinetics. Different tumors were seen, with uptakes varying across patients and lesions [43]. This study confirmed the feasibility of using radiolabeled liposomal drugs in a pretreatment phase to identify patients most likely to benefit from the therapy.

The demonstrated drawbacks of using liposomes for tumor imaging lie in their slow accumulation, which was demonstrated to take almost 24–48 h [4,44], and their nonspecific accumulation in the liver and spleen.

#### 3.2.3. VesCan

VesCan was a liposomal imaging agent made of highly pure phospholipids and cholesterol (mol ratio 2:1) loaded with ^111^In. It was developed by a South California consortium of the California Institute of Technology, the City of Hope, and Vestar Inc. (which later became NeXstar Pharmaceuticals, and was subsequently acquired by Gilead Sciences). This liposomal formulation reached phase III clinical trials but was never commercialized [45]. Studies first identified the best conditions for loading liposomal vesicles with ^111^In, which involved using nitrilotriacetic acid (NTA) inserted in the liposome as a chelator, and the ionophore A23187, which resulted in a loading of 80% of ^111^In. Then, VesCan was tested in vivo, where it identified tumor sites 24–48 h after injection. It also showed an accumulation in the liver and spleen. It was clearly stable in the blood, while a time-dependent degradation was seen in the tumor and liver. The particles were well tolerated, the radiation dose was in the range of conventional imaging techniques, and various tumors were identified. VesCan was never commercialized, however, despite these results and its administration to nearly 400 patients. The liposomal radiotracer’s ability to detect known tumors was ~70%, well below the goal of 85%, but its specificity exceeded 96%, with few false positive results. The production of VesCan was abandoned because of its limited ability to detect a tumor, and Vestar Inc. invested in developing therapeutic liposomes instead [46].

### 3.3. Silica NPs

Silica-based NPs have been widely used as drug delivery systems, largely thanks to the good biocompatibility of silica, an endogenous substance found mainly in bones. Silica NPs are “generally recognized as safe” by the FDA and have been used in clinical studies, too [47]. Based on their size, morphology, and composition, silica NPs (SiNPs) can be classified as dense (dSiO2), mesoporous (MSN) or biodegradable mesoporous (bMSN), and hollow mesoporous (HMSN) [47]. Then there are also the ultra-small core-shell SiNPs, called Cornell prime dots (C’ dots), which are NPs characterized by a silica core, a PEG shell, and a diameter of less than 8 nm [48].

As well as their biocompatibility, SiNPs have the advantage of simple surface engineering. Radiolabeled SiNPs can be further conjugated with targeting ligands to achieve the best conditions for molecular imaging. When functionalized with biomolecules, SiNPs are preferably radiolabeled after adding biomolecules and under mild conditions to avoid damaging them [47].

The drawbacks of SiNPs applications were always associated with the lack of a detention method enabling us to understand the absorption, distribution, metabolism, and excretion of these NPs. The opportunity to conjugate radioactive nuclides with an adequate half-life to these NPs nonetheless promoted their clinical translation until 2014, when a first-in-human clinical trial of ^124^I-labeled C’ dots was performed [47].

After this first-in-human trial, many other in vitro and in vivo studies were conducted using C’ dots to make them even more effective. For instance, the presence of a targeting agent on the surface of C’ dots may enhance their tumor penetration, accumulation, distribution, and retention (Table 3) [48,49].

### 3.4. Gold NPs 

Gold NPs are generally very small particles (<10 nm) with different shapes, such as spheres, rods, stars, and clusters. They are completely biocompatible and have a favorable biological half-life. They are non-toxic and can easily penetrate cells. They are readily modifiable using different chemical entities like chelators, targeting biomolecules or drugs [7,50], partly thanks to the presence of a negative charge on their surface [51]. Thanks to all these properties, gold NPs can deliver many molecules—including drugs, genes, and imaging agents—even when the latter have poor pharmacokinetic properties [7,51]. As gold NPs can be functionalized with targeting agents to enhance their tumor localization, they can also be useful for a more precise diagnosis and therapy. However, most studies have continued to focus more on the biodistribution and pharmacokinetic properties of AuNPs [50]. Authors of studies conducted on gold NPs have noted that, regardless of their size and shape, they may accumulate in organs like the liver, lung, and spleen [51].

Gold NPs can also be functionalized with radionuclides via different synthetic pathways: (1) using bifunctional molecules to stabilize the NP during the synthesis and bind the radioisotopes; (2) by directly conjugating molecules on the surface of AuNPs, especially those expressing amino and thiol groups; (3) via a ligand exchange of capping molecules on AuNPs with different molecules that have gold bonding capabilities; (4) by chemically modifying molecules already on the surface of the AuNPs [50]. In some cases, such as ^131^I and ^64^Cu, the radionuclide can be attached to the NP’s surface by absorption [50].

Radiolabeled gold NPs have been widely tested in pre-clinical applications for use in tumor imaging. Table 4 summarizes some pre-clinical studies involving radiolabeled gold NPs.

All these studies highlighted some features; for example, the presence of targeting agents on the NPs’ surface enhances their uptake in tumors, albeit with evidence of uptake in the adrenal glands as well as in the liver and spleen [52].

However, the possibility of degradation of the NPs’ shell is questioned by Kreyling and co-workers [54]. In their studies, they used ^198^Au NPs and modified their surface with ^111^In. They noted a significant difference in the retention of the two radioisotopes, which indicates a dissociation of the ^111^In shell from the core of the NP. This study thus revealed a potential disadvantage of using radiolabeled gold NPs.

By using ^64^Cu as a radionuclide, the risk of the dissociation of the tracer from the NP is reduced due to the embedding of the radionuclide inside the particle core. Furthermore, by using this radiolabelling approach, the surface of gold NPs is entirely available for functionalization with targeting agents or other agents like PEG, which was demonstrated to enhance the circulation time of ^64^Cu-AuNPs [55]. Even ^124^I can be used as a tracer for studying the biodistribution of gold NPs. It has a long half-life (4.17 days) and, like ^64^Cu it can be embedded at the core of the nanoparticles and also to the shell of the gold NPs [56].

More recently, gold nanoclusters (AuNCs) have also attracted great interest in radiotherapy. AuNCs are hybrid compounds with an inorganic metal core, consisting of 2–3 nm gold NPs and organic binders. They are organized in a structure ranging from 10 to 100 atoms with several advantages, such as high thermal conductivity, great optical stability, visible fluorescence, high stability, low toxicity, and low immune response. Nanoclusters can be made radioactive with thermal neutron irradiation, which makes them suitable for use in radionuclide therapy. Given their high potential, an in vitro possible application of the nanocluster named 198Au25 (Capt) 18 (which contains captopril as the ligand) was made. This study confirmed this nanocluster’s great potential efficacy for therapeutic purposes [57].

Like AuNPs, also AuNCs can be radiolabelled with ^64^Cu. The radionuclide is embedded in the structure of the AuNCs and easily controlled in the radiolabelling specific activity. Thus, a sensitive PET tracer can be obtained that is characterized by rapid clearance from the system, low accumulation in the RES, and a low nonspecific accumulation [53].

### 3.5. PLGA NPs

Polymer NPs can be made of natural or synthetic biodegradable polymers, such as polyanhydrides, polyethyleneimine, poly(lactic-co-glycolic acid) (PLGA), chitosan, and gelatine, and make up a huge group among the drug delivery vehicles [7]. Of all these polymers, PLGA is certainly one of the most studied, also for clinical applications. PLGA microspheres have been tested as long-acting drug delivery systems ever since 1984 and have been approved by the FDA for therapeutic applications (e.g., Lupron Depot) [58,59].

A variety of therapeutic agents can be encapsulated in PLGA NPs, including small molecules, proteins, and oligonucleotides. Inside the NP, they are protected against clearance and degradation, and their release is controlled via diffusion through the NP wall and PLGA degradation [59]. PLGA is hydrolyzed in vivo through the breakdown of the ester bonds and the action of tissue cells that phagocytize NPs, and hydrolyze them to produce lactic and glycolic acid; then, monomers are eliminated through the Krebs cycle [58]. In vivo, the PLGA degradation rate and release of the encapsulated molecule are influenced by different factors, including the molecular weight of the NP, the crystal profile, the storage temperature, and the surface coating materials. The lactate-to-glycolate ratio also has a major role in the particle degradation kinetics; lactic acid is more lipophilic than glycolic acid, so increasing amounts of lactic acid make the particle more lipophilic, and the release rate decreases [60]. The size of the NP influences the kinetics, too; small NPs are degraded faster in the spleen and liver than large ones [58]. The size and morphology of PLGA NPs depend on several synthetic parameters, such as the solvent used.

To obtain data on the pharmacokinetics and biodistribution of PLGA NPs in vivo, it has been essential, once again, to be able to radiolabel them. Radio-pharmacokinetic studies have mainly examined how molecules conjugated on the NP’s surface influence their distribution. Table 5 lists some pre-clinical studies conducted on radiolabeled PLGA NPs.

The presence of polysorbate-80 on the surface of PLGA NPs was found to reduce their uptake by phagocytic cells in the liver and spleen while enhancing brain uptake [61]. Even the presence of targeting moieties, i.e., folic acid and a peptide targeting vascular endothelial growth factor, on the surface of NPs was demonstrated to be useful for drug delivery purposes [62].

Monoclonal antibodies can be used for targeting purposes, too. For instance, the conjugation on the NPs’ surface of monoclonal antibodies against β-HCG was demonstrated to enhance tumor targeting and reduce renal uptake [63]. In the same study, even the presence of PEG was shown to influence the biodistribution of the NPs. While NPs with and without a PEG coating showed the advantages mentioned above, only the PEG-coated particles had a reduced liver uptake and the slowest blood clearance. [63] Moreover, PEG can also be used to bind the radionuclide like in the study of Sirianni and co-workers [59], where they conjugated a [^18^F]4-fluorobenzylamine to an NHS-PEG-biotin ([^18^ F]NBP4) to bind the radionuclide to avidin-modified PLGA NPs.

The NPs’ composition may influence their biodistribution, too. Comparing PLGA NPs with lactate-to-glycate ratios of 75:25 and 50:50, the latter showed a higher encapsulation efficiency (probably because they were less lipophilic) and had more favorable release kinetics. However, once the 50:50 PLGA NPs had been injected in vivo, the RES was confirmed as the main clearance route, although adding PEG on the particle’s surface helped to reduce RES uptake [60].

PLGA NPs were also tested to deliver radionuclides in a combined radio-chemotherapy. Gibbens-Bandala and co-workers [64] investigated in vivo the delivery of paclitaxel loaded in PLGA NPs radiolabeled with ^177^Lu and functionalized with bombesin as the targeting moiety. The results showed an enhanced drug release in vitro at pH 5.3 due to hydrolysis of the PLGA polymer, suggesting that this kind of NPs may improve drug delivery. A synergic effect of chemotherapy and radiotherapy on the cancer cells was also seen.

To our knowledge, none of the above-mentioned pre-clinical studies progressed to the clinical phase, despite the encouraging results.

### 3.6. Iron Oxide NPs

The ability of iron oxide NPs (IONPs) to act as both MRI contrast agents and PET tracers has recently made them highly attractive in nuclear medicine, especially since the advent of PET/MR scanners [65]. These particles usually have a magnetic core surrounded by a polymer or metal coating that can be functionalized with different types of molecules. They can be moved under the influence of a magnetic field, which can be used to bring them to the tumor site and to avoid their accumulation in healthy tissues [7].

Superparamagnetic iron oxide NPs (SPIONs) are of particular interest for their imaging applications. They are IONPs with a particle size of 5–25 nm, an appropriate surface coating, and an intrinsic superparamagnetism that gives them magnetic properties only in the presence of a magnetic field [66]. There are also ultra-small iron oxide (USPIO) nanoparticles, less than 5 nm in size, that have recently gained attention as promising MRI agents due to their magnetic properties, good tissue penetration, better biocompatibility, and longer blood half-life. They are partially metabolized by the kidneys, and their small dimensions make it easy for them to escape nonspecific uptake by the mononuclear phagocytic system [67].

To use these IONPs in PET/MRI, their surface could be labeled with radionuclides like ^99m^Tc, ^125^I, ^111^In, ^18^F, ^64^Cu, and many others. The most widely used radiolabeling strategy involves exogenous chelators because this enables the radionuclide to be added in the last stage of the synthesis. Chelators have their drawbacks, however, including complex coordination chemistry, the risk of altered pharmacokinetics, and the potential detachment of radioisotopes during imaging. A possible alternative is to use an intrinsic radiolabeling of the NP, which can be synthesized using methods such as hot-plus-cold precursors, specific trapping, cation exchange, and proton beam activation [65,68].

Many pre-clinical studies have been conducted on the use of IONPs for imaging purposes in recent times (Table 6).

Like other NPs, they have been conjugated with targeting moieties to improve tumor detection with successful results [67,69,72]. Two studies [66,68] examined the integrity and distribution of these NPs after their injection in vivo using radionuclides as tracers. Data collected by both studies demonstrated an intrinsic instability of the NPs after their in vivo injection, leading authors to recommend conducting stability studies before administering IONPs.

Alongside all these pharmacokinetic and biodistribution studies, research was also conducted on using IONPs to deliver drug therapeutics like doxorubicin [70]. The high cellular uptake and consequently high cytotoxicity of these NPs highlighted their potential for use both in PET/MRI and therapy. SPIONs radiolabeled with ^99m^Tc were also tested for sentinel lymph node detection by combining SPECT and MRI techniques [71]. The accumulation of radiolabeled SPIONs in lymph nodes paves the way for the future use of these nanoparticles, even in breast cancer and malignant melanoma.

To our knowledge, none of the pre-clinical studies on IONPs progressed to the clinical phase, despite the above-mentioned promising results. The combination of PET or SPECT with MRI can be advantageous in pre-clinical but mainly in clinical practice. There are several reasons: (1) MRI does not emit ionizing radiation; (2) MRI has an excellent temporal and soft-tissue resolution; (3) hybrid medical imaging can be done ins a single session, thus saving resources for hospital and time for the patients; (4) the opportunity to obtain spectroscopic and metabolic information from the radiolabeled molecules [73]. Therefore, in our opinion, the bimodal application would be interesting in the near future. However, a lot of barriers should be overpassed, such as investments, safety, and effectiveness.

### 3.7. Micelles

Micelles are formed by amphiphilic copolymers that self-assemble in an aqueous environment. They have a spherical structure with a hydrophobic core that can incorporate hydrophobic drugs and a hydrophilic shell providing steric stability. The corona can also be modified with vectors and/or imaging agents to enhance the specificity of the drug delivery. Micelles have excellent biocompatibility and can encapsulate a wide variety of drugs, and they preferentially accumulate in tumors via the EPR effect. That said, the nature of the copolymers used to prepare them and the surface charge, core-drug compatibility, drug loading capacity, drug release, and kinetics may affect the pharmacokinetic and distribution profile of the encapsulated drug [74,75].

Over the years, many formulations of micelles have been tested in vitro and in vivo, generating information on their pharmacokinetic and biodistribution properties, thanks in part to their labeling with radionuclides (Table 7).

Indeed, the presence of a radionuclide on the surface of micelles was demonstrated to also be an optimum strategy for testing the fate of micelles once “in vivo” injected [76]. Like liposomes, even micelles can be radiolabeled not only with the help of a chelator but also by a core-entrapment strategy. This strategy brings huge advantages, leaving the corona of micelles unaffected and preventing any interference by the chelator on the biodistribution and pharmacokinetic properties of micelles [78].

Micelles can also be loaded with chemotherapeutic agents. Micelles loaded with docetaxel showed a higher anti-proliferative effect than the docetaxel alone. Moreover, the presence of a radionuclide highlighted that the efficacy of micelles was due to their accumulation in cell lines [74]. Similarly, the radiolabeling of pH-sensitive doxorubicin-loaded micelles let to evaluate their higher accumulation in tumor sites compared to non-targeted tissues [75]. However, micelles can be radiolabeled also with a therapeutical radionuclide. Indeed, micelles simultaneously loaded with doxorubicin and ^188^Re were demonstrated to be more efficient in tumor treatment than the two therapeutic agents taken individually [77].

The studies all identified the accumulation of micelles in the spleen and liver, indicating the dose-limiting role of the RES-associated organs. All micelles seemed suitable for tumor targeting; however, achieving a high tumor accumulation largely thanks to the EPR effect [74,75,76,77,78].

### 3.8. Dendrimers

Dendrimers are branched polymeric structures originating from a core molecule [3,7]. They are highly symmetric and spherical and can be used to deliver hydrophobic compounds and anticancer drugs, which are encapsulated through physical interactions. Their physical and chemical properties can be modified to control their rate of degradation and the rate of the drug’s release as a consequence. There are four main factors governing dendrimer degradation: (1) the strength of the monomers’ chemical bonds; (2) the dendrimer’s hydrophobicity; (3) the chemical reactivity of the macromolecule; (4) the dendrimer’s molecular weight and generation number. Thanks to the EPR effect, dendrimers can deliver drugs to the tumor efficiently, so they have found numerous important applications in various fields, from gene antisense therapy to MRI [3]. The fact that dendrimers could be labeled with radionuclides led to their application in nuclear medicine, including SPECT, PET imaging, and radionuclide therapy. Because dendrimers could be functionalized simultaneously with imaging agents and chemotherapeutics, these delivery systems could also be used for multimodality imaging, theragnostic, and image-guided drug delivery. Table 8 summarizes some pre-clinical studies involving dendrimers.

One of the biases in dendrimer’s delivery may be represented by a rapid clearance, especially when the EPR effect is the strategy of choice for tumor targeting. Even in this case, PEG was tested to increase the circulation time of dendrimers, and the results were promising. Indeed, dendrimers labeled with PEG and ^99m^Tc circulated up to 24 h after injection [79].

However, the dendrimers’ delivery specificity can also be enhanced by adding targeting ligands to their surface [82]. For example, N. Song and co-workers [83] synthesized 131 I-labeled dendrimers suitable for SPECT imaging, radionuclide therapy, and antimetastatic therapy, thanks to the expression on their surface of a small homing peptide (LyP-1). Even folic acid was demonstrated to be a good targeting agent to enhance the delivery of dendrimers in the tumor site. Moreover, as demonstrated by Ma et al., the radiolabeling of dendrimers with ^64^Cu on dendrimers’ surface made them suitable for PET imaging [80] and probably also for therapeutical scope.

Although results obtained in pre-clinical studies were often encouraging, there was evidence of an excessive uptake of these NPs by the mononuclear phagocytic systems, which led to issues with the toxicity of these NPs [82].

## 4. Discussion

This review covers seven classes of nanoparticles that have been studied for the purpose of delivering radionuclides. There is clearly a huge gap between the numerous pre-clinical studies and their clinical translation. To our knowledge, micelles, dendrimers, iron oxide NPs and PLGA NPs have never been tested in clinical applications in nuclear medicine, even though some of them—like PLGA NPs—have FDA approval for clinical uses in other medical fields. This is probably due to problems associated with administering these classes of NPs. For instance, all the above-mentioned NPs have shown a nonspecific uptake by RES-associated organs, like the spleen and liver. This represents a huge drawback in nuclear medicine because it will give rise to high background signals during imaging, making diagnoses more difficult and causing toxicity in therapeutic applications. Indeed, when using a therapeutic radionuclide, its nonspecific uptake (i.e., physiological sites of biodistribution) will cause toxicity to some organs. Significant differences in uptake between pathological and healthy tissues are certainly a key feature of safe radiation therapy [84].

Radiolabeled gold and silica NP, on the other hand, have both been tested in humans [85,86], but studies only concerned a small or very small number of patients, so further investigations will be needed before any real application for medical purposes can be proposed.

These classes of NPs have also shown a nonspecific uptake by RES-associated organs, posing the same problems mentioned above for the other classes of NPs.

The biggest part of the studies focused on liposomes aims to understand their pharmacokinetic properties and how the surface elements can alter their biodistribution. Despite the other NPs, liposomes have been highly tested in patients for imaging purposes. A liposomal formulation named VesCan was the object of phase III clinical trials, but it was never commercialized. As more than one clinical study highlighted, a notable aspect of this class of NPs lies in that—along with the drawback of a nonspecific uptake by the liver and spleen—they offered no advantages over the known imaging techniques used in nuclear medicine. The specificity of their detection was significantly lower than was desirable and lower than that of modern molecular imaging.

However, the liposome-based clinical studies mentioned another possible application of this class of NPs, which probably represents the future for liposomes in imaging. Isotope-labeled liposomes could be used as probes to identify patients most suitable for treating with a specific liposomal formulation of “cold” conventional chemotherapeutics. A marked heterogeneity in NP uptake indeed came to light between different tumors in the same patient and between tumors of the same type in different patients, also making the answer to the therapy highly heterogeneous.

Because the heterogeneity in tumor uptake may belong not only to liposomes but also to the other types of NPs, the future for these delivery agents may be the same as for liposomes. Most of the NPs have already been studied for the delivery of therapeutic agents, and, as we have reported in the present descriptive review, all of them can be radiolabeled. Thus, hypothetically, the radiolabeled NPs can become probes to predict the response to therapy with the same NP caring for a chemotherapeutic (or even radiotherapeutic) agent.

Before electing NPs as the strategy of choice for the delivery of radionuclides, some problems should be solved, such as the nonspecific uptake of NPs by RES. The use of targeting agents may not represent the only solution to the problem because they add only a modest value in the avoidance of RES uptake of NPs [84]. Moreover, the use of PEG to prolong the circulation time of nanoparticles should be highly evaluated. Slow clearance both compromises contrast at early time points after administration and increase the patient’s exposure to the radiation. This is one of the reasons why antibodies are no more used as radionuclide delivery agents in nuclear medicine [84].

Thus, not only the delivery strategies but also the design of the NPs should be improved, for example, using pH-sensitive polymers or taking advantage of the presence of enzymes expressed on the tumor site and not in the healthy tissues) [84,87].

Once solved these problems, NPs must lastly demonstrate to be better than the already-used radionuclide-delivery agents before being applied in nuclear medicine practices.

## 5. Conclusions

In conclusion, NPs have been the object of many studies in nuclear medicine over the years, especially for imaging applications. We believe this field of research has great potential, even though a good deal of further investigations will be needed. However, their physiological accumulation in the liver and spleen should be solved before their application in therapy. Indeed, passive targeting and exploiting the EPR effect are still the main delivery pathways followed by NPs to reach their targets, but this makes the delivery scarcely specific and a far cry from the *magic bullet* concept. Moreover, more investigation on the possible toxicity of some kinds of NPs (i.e., iron oxide nanoparticles) is required. Finally, despite liposomes, a real translation of the pre-clinical setting in a clinical field hasn’t been done. In our opinion, some drawbacks still limit the step forward of the NPs mentioned above, mainly the costs and their final scope (drug delivery or radioisotope delivery).

Thus, we think that, based on the current state, radiolabeled NPs must be investigated longer before finding their place in nuclear medicine and, therefore, in clinical practice.

## Figures and Tables

**Figure 1 pharmaceutics-14-02024-f001:**
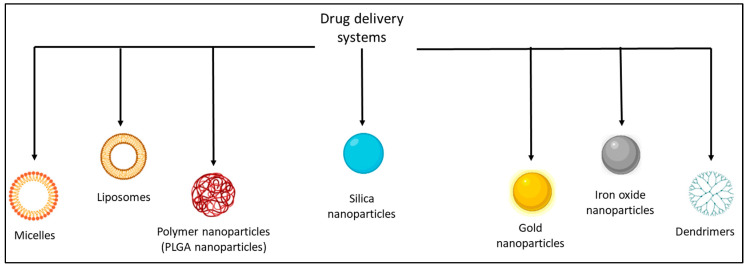
Drug delivery systems. This figure shows the main delivery systems analyzed in this review, with cartoons (made with the “BioRender” program) indicating the structure of each class of NPs.

**Figure 2 pharmaceutics-14-02024-f002:**
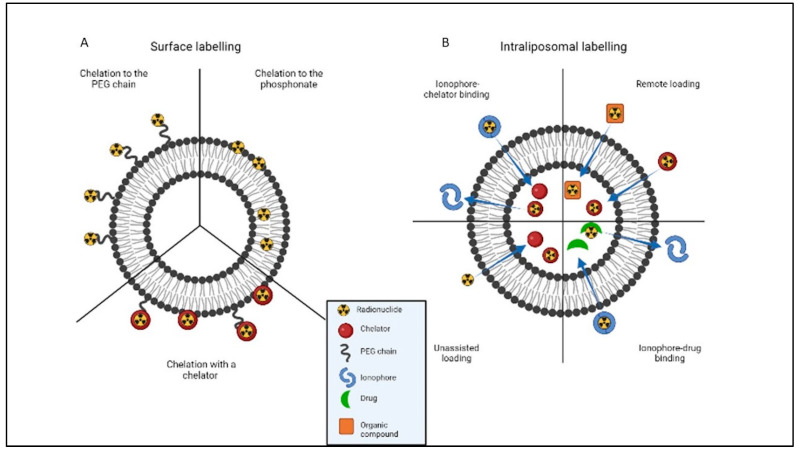
Liposome radiolabeling techniques. (**A**) Surface labeling can be done by radiolabeling the radionuclide directly to the phospholipid bilayer or a PEG chain, with or without the help of a chelator. (**B**) In intraliposomal labeling, the radionuclide is encapsulated in the core of the liposome. This can be done with or without the help of an ionophore. The binding may also require the presence of a chelator, or other drugs or organic compounds acting as chelating agents. Pictures made with the “BioRender” program.

**Table 1 pharmaceutics-14-02024-t001:** Pre-clinical studies on radiolabeled liposomes.

Reference	Year	Tracer	N° of Patients	Disease	Drug	Imaging Modality	Main Outcomes
Belhaj-Tayeb et al. [8]	2003	^99m^Tc	None	Tumor-bearing mice	None	Gamma camera	The uptake of ^99m^Tc-MIBI was enhanced when it was encapsulated in liposomes. Uptake of the liposomes by the spleen may be reduced with the pre-injection of cold liposomes.
Helbok et al. [35]	2012	^111^In	None	Tumor-bearing mice	None	Gamma camera	The use of targeting agents on the liposomes’ surface did not improve tumor uptake.
H.Lee et al. [33]	2015	^64^Cu	None	None	Doxil and MM-302 (doxorubicin)	PET/CT	Studies on the biodistribution of a radiolabeled liposomal formulation could be used to select the most suitable patients for such therapy.
Petersen et al. [36]	2015	^64^Cu e ^177^Lu	None	Tumor-bearing mice	None	PET/CT	The amount of PEG bound to the surface influences the liposomes’ biodistribution. PEGylation of the liposomes did not avoid their uptake by the liver and spleen. Liposomes may be simultaneously loaded with ^64^Cu and ^177^Lu to obtain a theragnostic agent.
Jansen et al. [34]	2018	^64^Cu e ^52^Mn	None	Tumor-bearing mice	None	PET/CT	Remote-loaded liposomes circulated longer than surface-labeled ones. Surface labeled liposomes may be unstable.
Alkandari et al. [37]	2020	^99m^Tc	None	None	None	Gamma camera	Positively charged liposomes were rapidly cleared from the blood and had a higher heart-to-liver ratio than uncharged liposomes.

**Table 2 pharmaceutics-14-02024-t002:** Clinical studies on radiolabeled liposomes.

Reference	Year	Tracer	N° of Patients	Disease	Drug	Imaging Modality	Main Outcomes
Richardson et al. [10]	1979	^99m^Tc	14	Different tumor types	None	Gamma camera	Negatively-charged liposomes were unable to localize in tumors. There was a high uptake of liposomes in the liver, heart, lung, and marrow.
Turner et al. Presant et al. [38,39]	1988	^111^In	24	Different tumor types	None	Gamma camera	The accumulation in tumor areas occurred 24 to 48 h after injection. The sensitivity of the detection was 85%, and the specificity 96%. There was liposome accumulation in the liver and spleen as well.
Koukourakis et al. [40]	1999	^99m^Tc	15	Different tumor types	Caelyx (doxorubicin)	SPECT/CT	Caelyx accumulated in tumor areas, the major thoracic vessel area, and the liver. Accumulation in tumors increased during the first 10 h.
Harrington et al. [41]	2001	^111^In	17	Different tumor types	None	Gamma camera	Liposomes accumulated in tumors, but also the liver and spleen. There was a very heterogeneous tumor uptake between different tumors, and between different patients with the same tumor.
Arrieta et al. [42]	2014	^99m^Tc	35	Malignant pleural mesothelioma	Doxorubicin	SPECT/CT	Patients with a ^99m^Tc-LD uptake of 75% or more showed a better response to the pharmacological treatment.
H. Lee et al. [43]	2017	^64^Cu	25	Breast cancer	MM-302 (doxorubicin)	PET/CT	The tumor was visible 2 to 3 days after the injection of radiolabeled MM-302 and to a variable degree between patients. Liposomes accumulated in the liver and spleen.

**Table 3 pharmaceutics-14-02024-t003:** Pre-clinical studies of radiolabeled silica NPs.

Reference	Year	Tracer	N° of Patients	Disease	Drug	Imaging Modality	Main Outcomes
Juthani et al. [49]	2020	^124^I, ^89^Zr	None	Mouse model of glioblastoma	None	PET	The PK of C dots was influenced by the presence of cRGD on their surface: tumor penetration was enhanced.
Zhang et al. [48]	2020	^177^Lu	None	Tumor-bearing mice	None	SPECT/CT	Labeling SiNPs with targeting agents enhanced their affinity for tumor cells.

**Table 4 pharmaceutics-14-02024-t004:** Pre-clinical studies of radiolabeled gold NPs.

Reference	Year	Tracer	N° of Patients	Disease	Drug	Imaging Modality	Main Outcomes
Ng et al. [52]	2014	^111^In	None	Glioblastoma model	None	SPECT/CT	Integrating the radionuclide in the NP left the whole surface free for functionalizing with the targeting agent. Uptake was seen in the tumor and the adrenal glands, liver, and spleen.
Zhao et al. [53]	2014	^64^Cu	None	Tumor-bearing mice	None	PET	The radiolabeling specific activity can be easily controlled. The clearance is fast, while the RES accumulation and the nonspecific tumor accumulation are low.
Kreyling et al. [54]	2015	^198^Au ^111^In	None	None	None	Gamma spectrometry system	The shell made of ^111^In became dissociated from the core of the NP.
Frellsen et al. [55]	2016	^64^Cu	None	Tumor-bearing mice	None	PET	The longer half-life was reached with a PEG coating compared to Tween-20-stabilized coating and a zwitterionic coating (sulfonic acid and quaternary ammines)
Pulagam et al. [56]	2019	^124^I	None	Tumor-bearing mice	None	PET	Gold NPs can be efficiently labeled with ^124^I both at the core and the shell. Poor accumulation of these NPs was seen in the tumor site.
Xuan et al. [57]	2020	^198^Au	None	None	None	Gamma spectrometry system	Nanoclusters have the potential to be very effective for therapeutic purposes: in all cell lines tested, they obtained better results in terms of cell death than low or medium doses of paclitaxel.

**Table 5 pharmaceutics-14-02024-t005:** Pre-clinical studies of radiolabeled polymeric PLGA NPs.

Reference	Year	Tracer	N° of Patients	Disease	Drug	Imaging Modality	Main Outcomes
Halder et al. [61]	2008	^99m^Tc	None	None	None	Gamma camera	Uptake in the liver and spleen was reduced after coating the PLGA NPs with polysorbate-80.
Arora et al. [60]	2012	^177^Lu	None	None	None	Gamma camera	PLGA NPs with a 50:50 ratio of lactate to glycolate had a higher encapsulation of ^177^Lu-DOTATE than NPs with a 75:25 ratio. Once injected in vivo, there was uptake of the PLGA NPs with the 50:50 ratio in the kidneys, liver, spleen, and gut. Adding PEG on the NPs’ surface reduced the RES uptake.
Sirianni et al. [59]	2014	^18^F	None	None	None	PET	A [^18^F]-fluorobenzylamine was bonded to an NSH-PEG-biotin, creating a ligand for functionalizing avidin-modified PLGA NPs.
He et al. [62]	2016	^99m^Tc	None	None	None	Gamma camera	Thanks to the presence of targeting agents, the NPs accumulated in the tumor. High retention in blood, liver, kidney, and bladder uptake was also seen.
Arora et al. [63]	2016	^177^Lu	None	Tumor bearing mice	None	Gamma camera	PLGA NPs functionalized with targeting agents and PEG were able to enhance tumor targeting and reduce renal and liver uptake and blood clearance.
Gibbens-Bandala et al. [64]	2019	^177^Lu	None	Tumor bearing mice	Paclitaxel	PET/CT	There was an enhanced drug release in vitro at pH 5.3. A synergic effect of chemotherapy and radiotherapy was observed. The greatest cytotoxic effect was between 24 and 72 h after injecting the NPs.

**Table 6 pharmaceutics-14-02024-t006:** Pre-clinical studies of radiolabeled iron oxide NPs.

Reference	Year	Tracer	N° of Patients	Disease	Drug	Imaging Modality	Main Outcomes
HY Lee et al. [69]	2008	^64^Cu	None	Tumor-bearing mice	None	PET/MRI	NPs showed a successful tumor delivery and also a high-RES uptake, probably because of their large size.
Yang et al. [70]	2011	^64^Cu	None	Tumor-bearing mice	Doxorubicin	PET/CT	Thanks also to the presence of targeting agents, there was a higher tumor uptake of NPs and a two-stage release of doxorubicin at pH 5.3. These NPs may pave the way for theragnostic applications.
Madru et al. [71]	2012	^99m^Tc	None	None	None	PET/MRI	SPIONs were able to accumulate in lymph nodes.
Wang et al. [66]	2015	^111^In ^59^Fe ^14^C	None	None	None	Gamma counter	Absolute stability of the NPs could not be demonstrated. Stability studies are always necessary before administering IONPs.
Sun et al. [67]	2019	^125^I	None	Tumor-bearing mice	None	SPECT/CT and MRI	A dimeric cRGD on the NPs’ shell enhanced their tumor accumulation and reduced the time they remained in non-targeted tissues, thus enhancing the tumor-to-background ratio. These nanoparticles seemed very good for SPECT and MRI.
Zhang et al. [68]	2019	^59^Fe ^64^Cu	None	None	None	PET	Directly radiolabeled NPs showed intrinsic instability after injection in vivo. A solution might be encapsulating the IONPs in a polymeric shell before functionalizing them.
Park et al. [72]	2020	^64^Cu	None	None	None	None	NPs were stable in buffer and human serum for 24 h. Uptake was good in breast, oral, and lung cancer cells.

**Table 7 pharmaceutics-14-02024-t007:** Pre-clinical studies of radiolabeled micelles.

Reference	Year	Tracer	N° of Patients	Disease	Drug	Imaging Modality	Main Outcomes
Kao et al. [76]	2013	^131^I	None	Tumor-bearing mice	None	Gamma camera	Despite RES uptake, radiolabeled micelles were demonstrated to accumulate highly in tumor sites thanks to the EPR effect
Shih et al. [77]	2015	^188^Re	None	Tumor-bearing mice	Doxorubicin	SPECT/CT	^188^Re and doxorubicin act synergically against tumor growth
Ribeiro et al. [74]	2016	^99m^Tc	None	None	Docetaxel	None	Micelles loaded with docetaxel showed higher anti-proliferative efficacy than the docetaxel alone. Micelles showed a long half-life
Laan et al. [78]	2016	^111^In	None	None	None	SPECT/CT	Micelles highly accumulated in the liver and spleen
Cavalcante et al. [75]	2021	^99m^Tc	None	Tumor-bearing mice	Doxorubicin	Gamma camera	Micelles loaded with doxorubicin can inhibit tumor growth better than doxorubicin alone. Nanosystems present low systemic toxicity

**Table 8 pharmaceutics-14-02024-t008:** Pre-clinical studies of radiolabeled dendrimers.

Reference	Year	Tracer	N° of Patients	Disease	Drug	Imaging Modality	Main Outcomes
McNelles et al. [79]	2015	^99m^Tc	None	Tumor-bearing mice	None	SPECT	PEGylated dendrimers are suitable for tumor delivery via the EPR effect due to their long circulation time (up to 24 h)
Ma et al. [80]	2018	^64^Cu	None	Tumor-bearing mice	None	PET	The presence of folic acid on the surface of dendrimers enhanced their specific targeting of the cell expressing folic receptor (in vitro) and to the tumor site (in vivo)
Song et al. [81]	2020	^131^I	None	Tumor-bearing mice	None	SPECT	^131^I-labeled dendrimers modified with LyP-1 were suitable for SPECT imaging, radionuclide therapy, and antimetastatic therapy

## Data Availability

Not applicable.

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
