# Peer review of "Nanoparticles and Radioisotopes: A Long Story in a Nutshell"

_pharmaceutics, 2022, doi:10.3390/pharmaceutics14102024_

Round 1

Reviewer 1 Report

The paper by Poletto et al., entitled “Nanoparticles And Radioisotopes: A Long Story In A Nutshell”, is a comprehensive (despite not exhaustive) review that collects and briefly discusses works reported in the literature concerning the use of radionuclides with different NP types, both with diagnostic and therapeutic purposes.

The outline of the manuscript is a bit confusing. It starts more as an “educational book chapter” and progresses to a “review style” diocument. Additionally, the weight that is provided to the different sections is unbalanced. Of course, it can be expected that the section collecting “liposomes” should be longer than others, as these nanoformulations have been more widely used in both preclinical and clinical scenarios… however, I find the sections “micelles” and “dendrimers” a bit underrepresented, with no specific examples. Could the authors include specific examples of their use? Also, in the section of gold NPs, the authors do not provide specific examples with PET imaging, only SPECT…  and there are in vivo works (preclinical) using 64Cu or 124I, for example.  Any reason not to include them?

Also, the authors mention in the “methods section” that “A literature search was conducted … No filters were used, apart from a minimum limit of 10 patients enrolled for clinical studies.” This criterium is as valid as any other and I have nothing to say… however, any limiting criterium for preclinical studies? Maybe a study with 3 human subjects is more relevant than a study with 10 mice…

Another general comment: the authors, in line 55, mention that “The aim of this narrative review … in order to answer the question: will NPs, for future, be the delivery system of choice for radionuclides?” But the authors do not answer the question. Of course, they give some clues about the limitations and advantages of NPs. However, a bit more discussion (and eventually own opinion) would be valuable. My main concerns are: 1- To prevent accumulation in the RES, NPs are functionalized with e.g. PEG to prolong circulation time … but long circulation time is not good for imaging purposes (low signal to noise, higher dose for the patient). This is one of the reasons why antibodies are not finding their way in diagnostics…  and 2- for (radio)therapeutic purposes, one needs to increase uptake in the tumour and decrease uptake in other sites. Isn’t RES accumulation preventing any therapeutic application of NPs? I fully agree in the statement “isotope-labeled liposomes could be used as probes to identify patients most suitable for treating with a specific liposomal formulation of "cold” conventional chemotherapeutics”. This can be extended to any NP… the authors may want to further elaborate on this. Maybe the authors should refer to a recent article (DOI: 10.1021/acsnano.1c09139) and take into consideration and discuss the aspects included there.

My final general comment: when the authors refer to the different labelling strategies for NPs, they usually include reviews as references. This is fine… although respecting the original works is also a good practice. I would combine both.   

Specific comments:     

Line 77: The authors claim that there are “There are three possible pathways by which NPs can reach a tumor site…” and they distinguish between passive accumulation via EPR effect and active targeting. This is not strictly correct. For active targeting, the first thing that NPs need to do is to reach the tumour site, and this happens via EPR, unless the functionalization aids in the active transport through blood vessels or the targeting moieties actually target vascular receptors. Hence, this sentence may need clarification. There is a general perception that targeting moieties help NPs to reach the tumour, and this is not always the case. Usually, targeting moieties prolong retention or enhance internalization in cells… but first NPs need to reach the tumour site.

Line 135: “already on the marker” should be “already in the market”.

Line 385: In the whole section of Iron oxide NPs, the authors suggest the advantages of dual PET/MRI (or SPECT/MRI) imaging. Is there any particular example in which the combination is powerful, or more powerful that the two modalities separated? I have the feeling that combining imaging modalities within one single probe is a technological (and scientific) challenge without practical application… discussion on this would be attractive.   

Author Response

The paper by Poletto et al., entitled “Nanoparticles and Radioisotopes: A Long Story In A Nutshell”, is a comprehensive (despite not exhaustive) review that collects and briefly discusses works reported in the literature concerning the use of radionuclides with different NP types, both with diagnostic and therapeutic purposes.

Q1. The outline of the manuscript is a bit confusing. It starts more as an “educational book chapter” and progresses to a “review style” document. Additionally, the weight that is provided to the different sections is unbalanced. Of course, it can be expected that the section collecting “liposomes” should be longer than others, as these nanoformulations have been more widely used in both preclinical and clinical scenarios… however, I find the sections “micelles” and “dendrimers” a bit underrepresented, with no specific examples. Could the authors include specific examples of their use? Also, in the section of gold NPs, the authors do not provide specific examples with PET imaging, only SPECT…  and there are in vivo works (preclinical) using 64Cu or 124I, for example.  Any reason not to include them?

R1. Micelles and dendrimers sections have been enriched, by including some sentences across all the text. Moreover, in the section of the gold nanoparticles some examples have been added on their radiolabeling with 64Cu and 124I, as suggested by the Reviewer.

Q2. Also, the authors mention in the “methods section” that “A literature search was conducted … No filters were used, apart from a minimum limit of 10 patients enrolled for clinical studies.” This criterium is as valid as any other and I have nothing to say… however, any limiting criterium for preclinical studies? Maybe a study with 3 human subjects is more relevant than a study with 10 mice…

R2. We have adopted the strategy of 10 patients for the clinical studies, to avoid including the first in human studies and for having “more consolidated clinical data”. This strategy is often used in systematic review for clinical endpoints.

Q3. Another general comment: the authors, in line 55, mention that “The aim of this narrative review … in order to answer the question: will NPs, for future, be the delivery system of choice for radionuclides?” But the authors do not answer the question. Of course, they give some clues about the limitations and advantages of NPs. However, a bit more discussion (and eventually own opinion) would be valuable. My main concerns are: 1- To prevent accumulation in the RES, NPs are functionalized with e.g. PEG to prolong circulation time … but long circulation time is not good for imaging purposes (low signal to noise, higher dose for the patient). This is one of the reasons why antibodies are not finding their way in diagnostics…  and 2- for (radio)therapeutic purposes, one needs to increase uptake in the tumour and decrease uptake in other sites. Isn’t RES accumulation preventing any therapeutic application of NPs? I fully agree in the statement “isotope-labeled liposomes could be used as probes to identify patients most suitable for treating with a specific liposomal formulation of "cold” conventional chemotherapeutics”. This can be extended to any NP… the authors may want to further elaborate on this. Maybe the authors should refer to a recent article (DOI: 10.1021/acsnano.1c09139) and take into consideration and discuss the aspects included there.

R3. The discussion has been modified, considering all the precious suggestions given by the reviewer.

Q4. My final general comment: when the authors refer to the different labelling strategies for NPs, they usually include reviews as references. This is fine… although respecting the original works is also a good practice. I would combine both. 

R4. References have been added.

Specific comments:     

Q5. Line 77: The authors claim that there are “There are three possible pathways by which NPs can reach a tumor site…” and they distinguish between passive accumulation via EPR effect and active targeting. This is not strictly correct. For active targeting, the first thing that NPs need to do is to reach the tumour site, and this happens via EPR, unless the functionalization aids in the active transport through blood vessels or the targeting moieties actually target vascular receptors. Hence, this sentence may need clarification. There is a general perception that targeting moieties help NPs to reach the tumour, and this is not always the case. Usually, targeting moieties prolong retention or enhance internalization in cells… but first NPs need to reach the tumour site.

R5. A clarification has been made.

Q6. Line 135: “already on the marker” should be “already in the market”.

R6. The term has been changed.

Q7. Line 385: In the whole section of Iron oxide NPs, the authors suggest the advantages of dual PET/MRI (or SPECT/MRI) imaging. Is there any particular example in which the combination is powerful, or more powerful that the two modalities separated? I have the feeling that combining imaging modalities within one single probe is a technological (and scientific) challenge without practical application… discussion on this would be attractive.   

R7. Some discussion about the advantages of using hybrid imaging systems with MRI and SPECT or PET has been included in the appropriate section.

Reviewer 2 Report

The authors review several types of nanoparticles, including liposomes, silica NPs, polymer NP s, gold NPs, etc.; discuss the results of preclinical studies and, in some cases, experiences from the clinic. They conclude that NPs are an attractive transport system, but have not yet been used in the practice of nuclear medicine.

The article succinctly organises the knowledge of nanoparticles in combination with isotopes and, in my opinion, can be a useful resource.

Author Response

The authors are thanksful to the reviewer for his/her comments.

Some comments have been included in the new version of the manuscript in accordance with the other reviewers.

Reviewer 3 Report

The authors report on a review about the use of nanoplatforms to deliver radiation in preclinical and clinical settings. This is a wide field with several reviews over the last decade however the authors defined the conditions for such a narrative.

The introduction is very concise about the use of well-established NPs for radiation delivery. Furthermore, the mechanism of action for tissue targeting is also highlighted. The authors present several examples in the last decade of preclinical and clinical studies for each nanoconstruct using different radionuclides.

The discussion and conclusion are well written where it is summarized the goal of most studies using NPs as well as their limitations to entering the clinical setting.

The authors also comment on the potential of using radiation to validate the pharmacokinetics and pharmacodynamics of novel nanoplatforms for diagnostic and therapeutics.

Author Response

(The authors gave the same response as above.)

Reviewer 4 Report

Authors need to be more focus on particles in nano-size not particles out of that size.

More literature survey is required.

Conclusion must be focused on the main topic of the review.

Author Response

Q1. Authors need to be more focus on particles in nano-size not particles out of that size.

R1.The present review was focalized on the nanometric particles.

Q2. More literature survey is required.

R2. Literature has been enriched also in accordance with the Reviewer 1.

Q3. Conclusion must be focused on the main topic of the review.

R3. The discussion has been implemented with opinions about the main topic of the review. Also a small sentence has been added in the conclusion section.

Round 2

Reviewer 1 Report

The authors have addressed all comments and concerns raised by the reviewer. The work can be published now. Despite I am not a native English speaker, I think that some proof-reading might be required before final publication.  

Reviewer 4 Report

The authors had promoted the manuscript.

It is accepted in current form